# Towards High-Efficiency Photon Trapping in Thin-Film Perovskite Solar Cells Using Etched Fractal Metadevices

**DOI:** 10.3390/ma16113934

**Published:** 2023-05-24

**Authors:** Ana Bărar, Stephen Akwei Maclean, Octavian Dănilă, André D. Taylor

**Affiliations:** 1Electronic Technology and Reliability Department, Polytechnic University of Bucharest, 060082 Bucharest, Romania; 2Chemical Engineering Department, Tandon School of Engineering, New York University, Brooklyn, NY 11201, USAandre.taylor@nyu.edu (A.D.T.); 3Physics Department, Polytechnic University of Bucharest, 060082 Bucharest, Romania

**Keywords:** fractal metadevices, perovskite solar cells, light trapping structures, electromagnetic field control

## Abstract

Reflective loss is one of the main factors contributing to power conversion efficiency limitation in thin-film perovskite solar cells. This issue has been tackled through several approaches, such as anti-reflective coatings, surface texturing, or superficial light-trapping metastructures. We report detailed simulation-based investigations on the photon trapping capabilities of a standard Methylammonium Lead Iodide (MAPbI${}_{3}$) solar cell, with its top layer conveniently designed as a fractal metadevice, to reach a reflection value R<0.1 in the visible domain. Our results show that, under certain architecture configurations, reflection values below 0.1 are obtained throughout the visible domain. This represents a net improvement when compared to the 0.25 reflection yielded by a reference MAPbI${}_{3}$ having a plane surface, under identical simulation conditions. We also present the minimum architectural requirements of the metadevice by comparing it to simpler structures of the same family and performing a comparative study. Furthermore, the designed metadevice presents low power dissipation and exhibits approximately similar behavior regardless of the incident polarization angle. As a result, the proposed system is a viable candidate for being a standard requirement in obtaining high-efficiency perovskite solar cells.

## 1. Introduction

Due to their impressive power-conversion efficiency values, reaching as high as 24.3% [1,2,3,4], as well as their simple and cost-effective fabrication process, perovskite solar cells represent one of the main areas of research in the field of photovoltaics. Furthermore, the ability to solution-process the precursor and obtain perovskite films through solution deposition has opened the possibility of developing highly efficient perovskite thin-film solar devices. This avenue has been extensively investigated, with promising results [5,6,7]. However, thin-film perovskite solar devices still have significant shortcomings, such as high thermally induced chemical instability [8,9,10,11], and current-voltage hysteresis [12,13]. Thermally induced chemical degradation is due to perovskite’s low formation energy [14,15], which can be easily reached at temperatures as low as 85 ${}^{\circ }$C [14,15], while hysteretic behavior of perovskite solar cells is caused by ion migration, charge carrier trapping/detrapping processes, and ferroelectric polarization of perovskite materials [16]. Defect passivation has been the main approach to mitigating ion migration and hysteresis reduction [17,18,19,20,21]. One of the most significant limitations of thin-film perovskite solar cells are caused by optical losses. Such losses are induced by optical phenomena occurring at the incidence of light on the illuminated surface of the cell, namely light reflection (reflective losses), transmission (transmission losses), and absorption (parasitic absorption losses). Parasitic absorption losses refer to light absorption in the perovskite layer which leads to heat dissipation within the device, instead of electron-hole generation, and transmission losses consist of light passing through the perovskite layer without being absorbed. Both types of losses are strongly linked to the perovskite layer thickness, as well as the perovskite’s optical properties. Reflection losses occur due the blocking of light at the illuminated interface of the cell, and they heavily depend on the refractive indices of the materials creating the interface, as well as their optical properties. Several approaches have been reported for tackling reflective losses in perovskite thin-film devices, such as anti-reflective coatings [22,23], surface texturing [24,25,26], light-trapping nanoparticles [27,28,29,30]. Metasurfaces are 2D, artificial structures, obtained by repeating unit elements, which exhibit unique electromagnetic properties [31]. These properties can be tuned by changing both the materials of which the structure is made, as well as the geometry and size of its unit elements. Usually, in order for a metasurface to respond a specific wavelength λ, the size of its unit element should be around one tenth of that wavelength $\left(\frac{\lambda }{10}\right)$ [32]. Such control over a structure’s properties, along with recent advances in micro and nanofabrication technologies [33,34,35,36], has allowed the design and fabrication of metasurfaces with specific electromagnetic properties, tailored for a wide range of applications, in specific ranges of the electromagnetic spectrum. For GHz and THz domains, metasurface designs have been demonstrated for applications such as imaging [37,38], wave front shaping [39,40], high-speed communication [41,42], and sensing [43,44]. In the visible domain, metasurfaces have been designed and tested as devices for beam shaping [45,46,47,48], metalenses [49,50,51,52], and light-trapping structures for thin-film solar cells [53,54,55,56]. Furthermore, perfect or near-perfect absorbing metasurfaces have been demonstrated for most domains, including visible [35,57], mid- and near-infrared [58,59,60], and GHz [61,62,63], with additional properties of frequency and polarization selectivity. Such designs have been proven suitable for applications such as solar energy harvesting [64,65,66,67], thermal detection [68,69], and biosesning [70,71,72].

In this paper, we report the design and theoretical investigation of several fractal metastructure, consisting of Methylammonium Lead Iodide (MAPbI${}_{3}$) structures obtained through several iterations of the same fractal patterning method, with two different structure profiles. The spectral behavior of the designed structures are studied under incident electromagnetic radiation in the ultra-violet (UV) and visible (Vis) range, specifically 0.27–0.8μm$(1.11\cdot 10^{3}$–375THz). The reflection spectra as functions of wavelength are extracted for all structures, in the interest of identifying the metadevice designs exhibiting anti-reflective behavior. One such structure is identified and subjected to further investigations regarding its power dissipation efficiency and spectral response to incident light polarization variation. The simulated results indicate that the metadevice has a low power dissipation ratio, and it is mostly invariant to changes in light polarization.

## 2. Design Considerations and Simulation Conditions

The standard p-i-n architecture of a perovskite solar cell is shown in Figure 1. It consists of a metallic electrode (usually silver or gold), an electron transporting layer (ETL), usually Tin Oxide $\left(SnO_{2}\right)$, followed by the perovskite active layer $\left(MAPbI_{3}\right)$, a hole transporting layer (HTL), usually 2,2${}^{\prime }$,7,7${}^{\prime }$-Tetrakis[N,N-di(4-methoxyphenyl)amino]-9,9${}^{\prime }$-spirobifluorene (Spiro-OMeTAD), and an optically transparent electrode, usually Indium Tin Oxide (ITO). Since this work is focused on the effects occurring at the illuminated interface, it is sufficient to consider the interface between $MAPbI_{3}$ and Spiro-OMeTAD for the metadevice design. For this reason, our metadevice contains the structure directly etched on the surface of the $MAPbI_{3}$ layer, which is covered with a Spiro-OMeTAD layer. In order to obtain the fractal geometry, a “C” element is considered as a generator and the sides of the element are considered as initiators. For each iteration, a “C” element of smaller size is repeated on each side of every existing “C” element, as shown in Figure 2. In order to obtain an optimal, low-level reflection structure, three patterns of increasing complexity are considered: “1C”, which only consists of the main “C” element, “2C”, obtained after one iteration of the fractal patterning, and “3C”, obtained after two iterations of the fractal patterning. Furthermore, based on the work of Baryshnikova et al. [27], where a low-level reflection spectrum in the visible range is demonstrated for a methylammonium lead tribromide $\left(MAPbBr_{3}\right)$ metasurface formed of rounded, needle-shaped elements, this study also considers two different structure profiles: a square-shaped profile, and a rounded, semi-elliptical element profile, where the element height $h_{e}$ represents half of the ellipse’s major axis (as shown in Figure 2), and the element widths $w_{1}$, $w_{2}$, and $w_{3}$ are the minor axis of the profiles for the main, medium, and small “C” elements. Also based on the work of Baryshnikova et al. [27], who have chosen an initial element height of 0.17μm, for this work, the element height $h_{e}$ starting value was chosen as 0.2μm. The sizes and spacings of the metadevice are given in Figure 2 and Table 1. Both the substrates and the fractal structures are made of $MAPbI_{3}$, with a top layer of Spiro-OMeTAD. The electric parameters of both $MAPbI_{3}$ and Spiro-OMeTAD are given in Table 2. Both materials are non-magnetic, with a magnetic permeability $\mu_{r}=1$.

The considered structures are simulated and investigated using a commercially available Finite Element Time Domain (FETD)-based simulation software (Comsol Multiphysics). The spectral behavior of the metastructures are investigated in the ultra-violet (UV) and visible (Vis) domains of the electromagnetic spectrum. The reflection spectra as functions of wavelength $R\left(\lambda \right)$ are determined for all considered structures, based on the values of the $S_{11}$ parameters of the structure’s scattering matrix, with the following relation:(1)$$R={\left|S_{11}\right|}^{2}$$

The simulated input electric field is limited to 0.27–0.8 μm
$(1.11\cdot 10^{3}$–375THz), which corresponds to the UV-Vis portion of the electromagnetic spectrum, with an incident power of $P_{i}=10^{-15}\;W$. This value corresponds to an effective intensity $I=0.25\;\mu W/{\mathrm{cm}}^{2}$ over the unit cell, which is high enough to observe significant photon trapping behavior, and low enough to cancel interference, multi-order diffraction and higher-order non-linearity effects such as Kerr lens modulation or multi-photon absorption. In order to control the polarization, the input electric field *E* is written in the form of the Jones vector:(2)$$\left(\begin{array}{c} E_{x}\\ E_{y}\\ E_{z} \end{array}\right)=\left(\begin{array}{c} \cos\left(\phi \right)\\ \sin\left(\phi \right)\cdot \exp\left(j\beta \right)\\ 0 \end{array}\right)$$
where ϕ is the linear polarization angle, and β is the elliptical polarization angle. In order to ensure the structure’s periodicity, the side walls of the block were assigned Floquet periodicity conditions for both electric and magnetic fields. Also, to account for the large thickness of the perovskite layer, the base of the substrate was simulated as an impedance matching condition for the incoming radiation, thereby canceling any possible transmission through the perovskite layer. The meshing of the architecture was designed such that the distance between two nodes ranges from $\lambda_{min}/5$ in the low resolution area (air and bulk perovskite propagation) to $\lambda_{min}/50$ in the high resolution area, represented by the “C” elements and the skin-depth of the interface between the “C” elements and the bulk perovskite.

## 3. Results and Discussion

The reflection response of the proposed structures are shown in Figure 3, along with the reflection spectrum simulated for a reference plane $MAPbI_{3}$ surface, with a top layer of Spiro-OMeTAD (shown in Figure 4). In order to have a high-efficiency photon trapping mechanism, the value of reflection must stay below 0.1 for all wavelengths within the visible range (0.4–0.8μm). All structures simulated with a square profile present reflection peaks well above 0.1, especially in the 0.65–0.8μm, regardless of structure complexity or element height. However, structure “3C”-square profile yields a more promising reflection spectrum for an element height $h_{e}=0.3\;\mu m$, maintaining reflection values below 0.1, except for three peaks: 0.13, at λ=0.418μm, 0.109 at λ=0.516μm, and R=0.125 at λ=0.695μm. These undesired peaks are successfully eliminated by rounding the profile of the structure. For the same element height $h_{e}=0.3\;\mu m$, structure “3C” with a rounded profile presents a reflection spectrum consistently below 0.1, throughout the visible range 0.45–0.8μm. The spectrum of this structure also presents zero reflection for three particular wavelengths: 0.448μm, 0.503μm, and 0.668μm, which implies that for these wavelengths, all radiation is absorbed. A rounded profile also improves the reflection spectra of structures 1C and 2C, greatly reducing the peaks found in the 0.4–0.65μm range. However, it does not succeed to completely attenuate the high reflection present in the 0.65–0.8μm range. Therefore, out of all the metadevice designs considered in this work, the “3C” structure, with a rounded profile and element height $h_{e}=0.3\;\mu m$, is the optimal geometry in reaching and maintaining a reflection value below 0.1, over the entire 0.4–0.8μm spectrum, and it will remain the focus of the remainder of this work. Baryshnikova et al. [27] reported reflection values R<0.04 in the visible spectrum for a $MAPbBr_{3}$ metasurface, with a glass substrate, immersed in a medium with refractive index 1.51. We have used their work as a starting point, however, we have taken into consideration several structural modifications which would allow our metadevice to model the illuminated interface of a p-i-n perovskite solar cell, such as a bulk perovskite active layer as substrate, and a Spiro-OMeTAD top layer ($n_{Spiro}=1.81\;$ [77]).

To ensure optimal photoelectric conversion in the bulk material, the heat conversion of the photons trapped by the proposed architecture must be minimal at all frequencies. Therefore, a further study is carried out on structure “3C”-rounded profile, concerning its power dissipation efficiency. Power dissipation in the active semiconducting layer of a solar cells introduce two parasitic effects: The first is the decrease of available photons for conversion, which effectively translates in an attenuation of the incoming field. The second is the conversion to heat by means of lattice vibrations inside the structure, which cause thermal agitation of free charge carriers, leading to an increase in charge carrier recombination rate and an overall decrease of the solar device’s efficiency. Due to the fact that the majority of the photoelectric interaction occurs at the interface (following a Beer-Lambert attenuation law), the study was only concerned with power dissipation along the skin-depth of the interface, and did not factor in bulk effects such as bulk heat conversion or net heat flow. Also, since the materials used are semiconductors, dilation effects of the elements, which can have a great deal of influence on the response of metallic structures are also rendered negligible. In order to verify the power dissipation efficiency, the total dissipated power density of the structure, $P_{d}\;\left(W/m^{3}\right)$, was calculated based on the simulated data, with the following relation:(3)$$P_{d}=\frac{1}{2}\cdot \sigma \cdot {\left|E\right|}^{2}$$
where σ is the electrical conductivity of the material, and *E* is the electric field strength. The ratio between the total dissipated power density $P_{d}$ and the power density $P_{i}$ incident on the structure is plotted as a function of wavelength, as shown in Figure 5. The total power dissipation density profiles of the structures correspond to the wavelengths at which the geometry yields zero reflection. The highest $\frac{P_{d}}{P_{i}}$ value is 0.122, yielded in the narrow wavelength range of 0.511–0.524 In the 0.27–0.8μm wavelength range, the average power dissipation ratio of the metadevice is $\langle \frac{P_{d}}{P_{i}}\rangle =0.0866$, while in the visible range (0.4–0.8μm), $\langle \frac{P_{d}}{P_{i}}\rangle$ is equal to 0.108 Such low power dissipation ratios indicate that the structure efficiently absorbs incident radiation for free charge carrier generation.

The spectral response of structure 3C/rounded profile to polarized light was also investigated, in order to verify whether the metadevice remains invariant to light polarization or not. The geometry’s reflection spectrum as a function of wavelength was simulated, for different values of the incident light polarization angle $\phi \;\left({}^{\circ }\right)$. The results are shown in Figure 6. Reflection values stay below 0.1, for all polarization angles ϕ, in the wavelength range of 0.45–0.65μm, except for a small peak of 0.107 at λ=0.423μm, for $\phi =0^{\circ }$. In the wavelength range of 0.65–0.8μm, reflection values increase over 0.1, however this increase does not surpass 0.13 This indicates that the metadevice is suitable for harvesting solar radiation, due to the fact that its reflection spectrum will remain largely unchanged, regardless of variations in incident light polarization.

As a last remark, we remind that in terms of sensitivity to construction errors, the element geometry provides a stable variational response: Any modification on the sizes and relative positioning of the fractal elements leads to a negligible frequency shift in the peaks, with the metadevice exhibiting less than 10% relative spectrum modification within a 5% tolerance limit applied to the variation of construction parameters.

## 4. Conclusions

In this paper, we investigate the photon trapping capabilities of a fractal metadevice, under electromagnetic radiation in the visible range (0.4–0.8μm). The study was performed under the criteria of low-reflection, low heat-conversion through power dissipation, and low sensitivity to input field polarization. The fractal structure is obtained after two iterations of the fractal patterning method, where a “C” element is considered as a generator, and the sides of the element are considered as intiators. Each iteration consists of repeating a “C” element of smaller size on each side of every already existing “C” element. The resulting elements have a rounded profile and an element height of 0.3μm. The substrate and the fractal structure are made of $MAPbI_{3}$, with a top layer of Spiro-OMeTAD. The spectral response of the structure yields reflection values consistently under 0.1, in the 0.4–0.8μm wavelength range, which indicates an average 0.15 decrease in reflection, compared to the reflection spectrum a simulated reference plane $MAPbI_{3}$ surface with a Spiro-OMeTAD top layer. Furthermore, the simulated metadevice an average dissipated power density to incident power density ratio of 0.108 in the visible range (0.4–0.8μm), which indicates that the structure exhibits low absorption losses. Further investigations show low sensitivity to variations in incident light polarization, the only notable increase in reflection caused by polarization angle variation being observed in the wavelength range of 0.65–0.8μm. However, this increase does not exceed 0.13, while the remainder of the reflection spectrum stays consistently under 0.1. Consistent low reflection values across the visible spectrum, low power dissipation rates, and limited responsiveness to light polarization variation render the proposed metadevice suitable to be used as a non-reflective surface for perovskite solar cells, with the purpose of increasing the net photoelectric conversion efficiency while maintaining low heat conversion capabilities.

## Figures and Tables

**Figure 1 materials-16-03934-f001:**
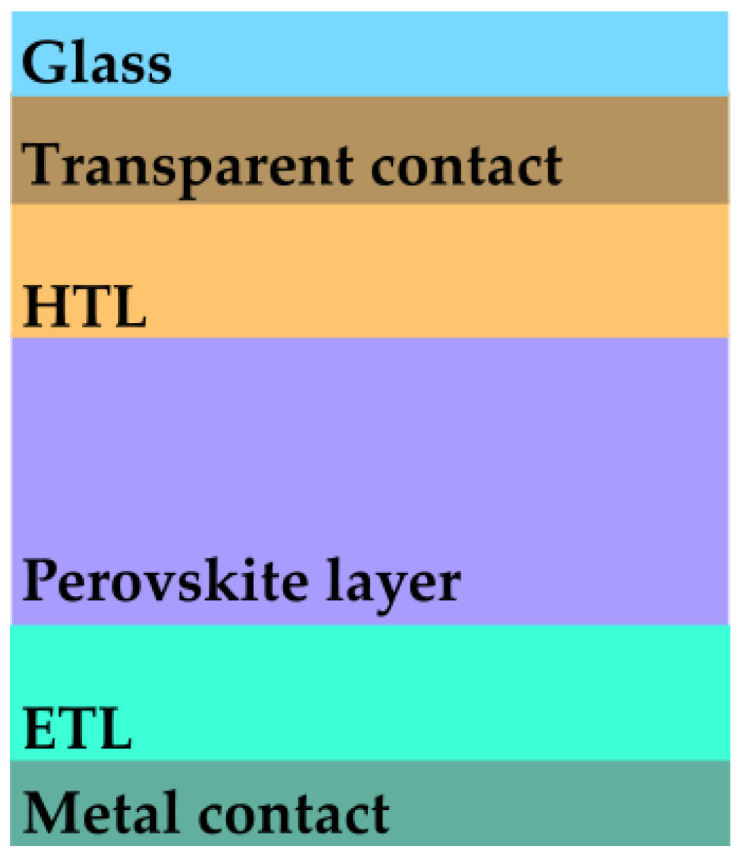
Perovskite solar cell p-i-n architecture, where ETL—electron transport layer, and HTL—hole transport layer.

**Figure 2 materials-16-03934-f002:**
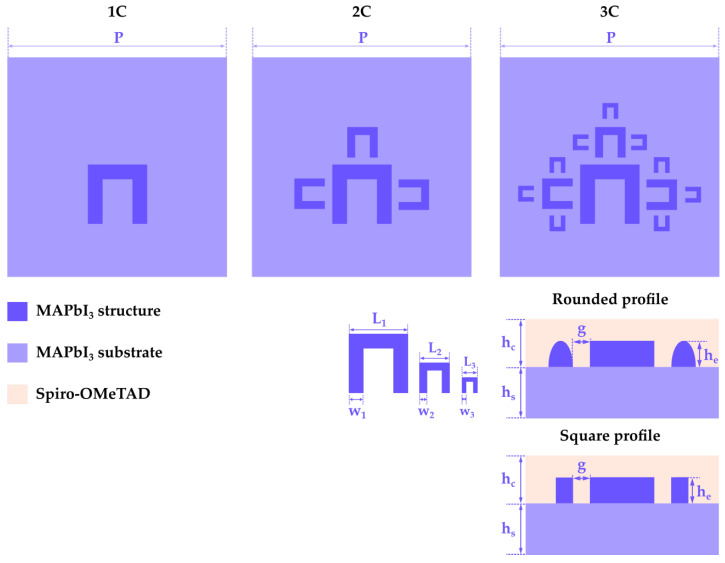
Layout and geometric parameters of the etched fractal metadevice design.

**Figure 3 materials-16-03934-f003:**
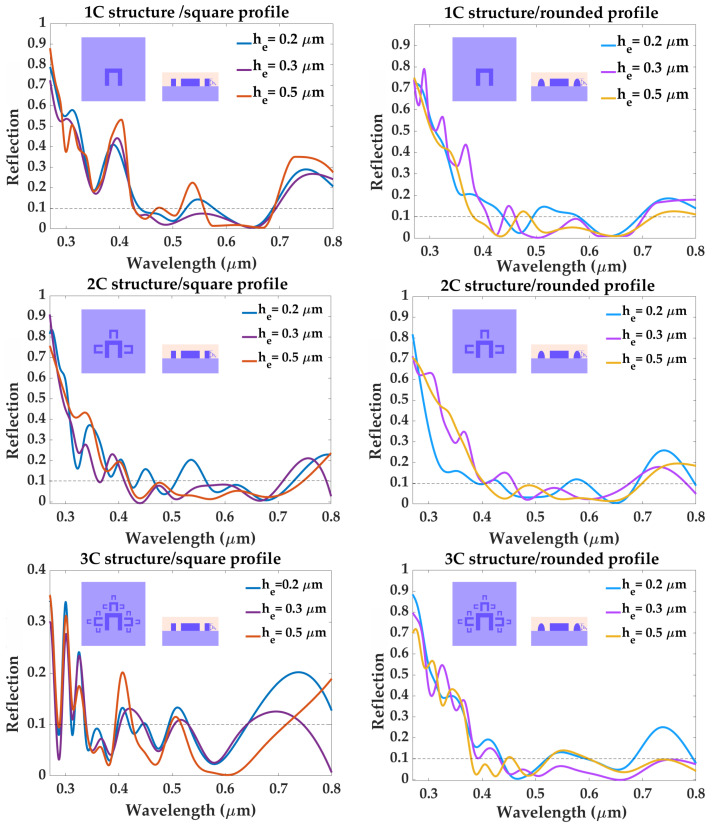
Reflection response of structures 1C, 2C, and 3C, respectively, in the wavelength range of 0.27–0.8μm. Reflection values become significantly lower than the reference cell as the complexity of the architecture and the roundness of the profile increase.

**Figure 4 materials-16-03934-f004:**
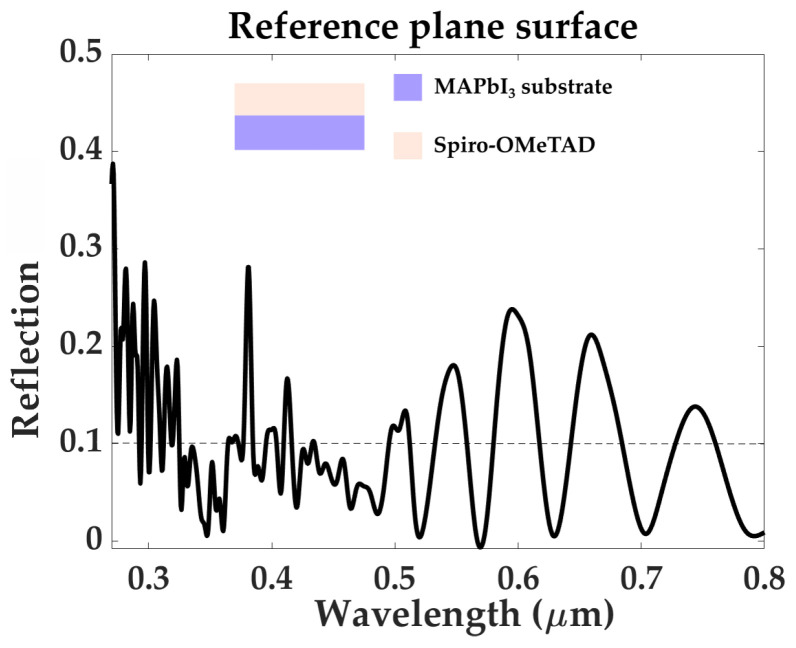
Reflection response of the reference plane surface cell, showing considerable reflection bands greater than 0.1 and up to 0.25 in many portions of the visible spectrum.

**Figure 5 materials-16-03934-f005:**
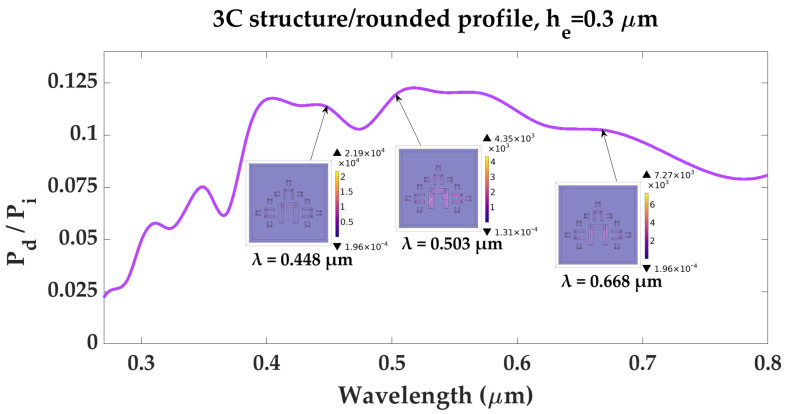
Power dissipation density ratio as a function of wavelength, for structure “3C”—rounded profile, with element height $h_{e}=0.3\;\mu m$, and total dissipated power density profiles at wavelengths 0.448μm, 0.503μm, and 0.668μm, where the structure presents zero reflection.

**Figure 6 materials-16-03934-f006:**
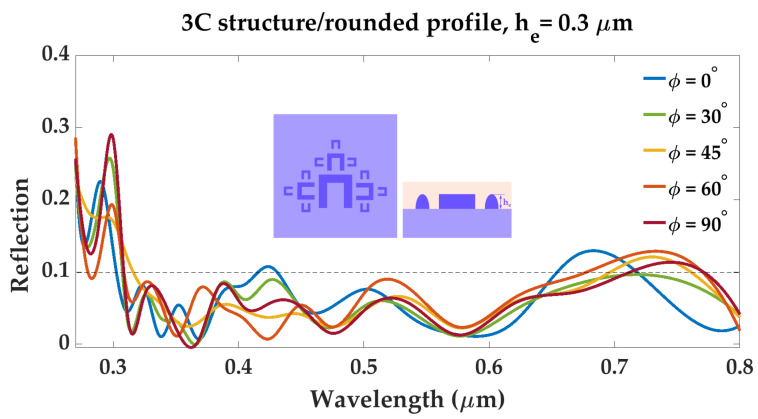
Reflection spectrum of structure “3C”—rounded profile, with element height $h_{e}=0.3\;\mu m$, for different polarization angles ϕ of incident radiation.

**Table 1 materials-16-03934-t001:** Metadevice structure geometric parameters values.

Element	Notation	Size (μm)
Main “C” element length	$L_{1}$	0.5
Main “C” element width	$w_{1}$	0.125
Medium “C” element length	$L_{2}$	0.25
Medium “C” element width	$w_{2}$	0.0625
Small “C” element length	$L_{3}$	0.125
Small “C” element width	$w_{3}$	0.03625
Substrate height	$h_{s}$	2
“C” elements height	$h_{e}$	0.2
Coating layer height	$h_{c}$	2
Gap between “C” elements	*g*	0.1
Cell size	*P*	2

**Table 2 materials-16-03934-t002:** Metadevice structure material properties.

Material	σ (S/m)	$\epsilon_{r}$	Reference
$MAPbI_{3}$	$3.17\cdot 10^{-4}$	7.5−j·0.02	[73,74]
Spiro-OMeTAD	$1.23\cdot 10^{-2}$	3	[75,76]

## Data Availability

Not applicable.

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
