# Peer review of "Towards High-Efficiency Photon Trapping in Thin-Film Perovskite Solar Cells Using Etched Fractal Metadevices"

_materials, 2023, doi:10.3390/ma16113934_

Round 1
Reviewer 1 Report
Comments on the manuscript
This manuscript proposes using a fractal metasurface as the top layer of a MAPbI solar cell to reduce reflective loss. Simulation-based investigations show that this approach can improve reflection values in the visible domain to less than 0.1. The suggested metasurface consumes less power and behaves consistently regardless of the polarization angle of the incoming light. These properties make it suitable for use in perovskite solar cells that aim for high efficiency.
The paper is scientifically intriguing, and the theoretical findings contribute to solar cell metasurfaces. Additionally, the manuscript is technically sound, with well-supported conclusions and assertions. Therefore, the manuscript is within the scope of Materials, and I recommend publishing it, provided that the minor points are addressed.
1. “Usually, in order for a metasurface to respond a specific wavelength λ, the size of its unit element must finds itself in the value range of (λ/10 ; λ/5)”. This claim is problematic since metasurface unit cells often require sub-λ. Thus, please be careful about your claim.
2. “For GHz and THz domains, metasurface designs have been demonstrated for applications such as imaging [26,27], wave front shaping [28,29], high-speed communication [30,31], and sensing [32,33].” The ability to absorb light perfectly is a crucial feature of metasurfaces in the mid-IR, near-IR, and visible ranges, which is highly related to this work. I suggest the authors discuss recent advancements in this area to give readers a complete understanding. Here are some references [Liang, Yao, et al. "Full-stokes polarization perfect absorption with diatomic metasurfaces." Nano Letters 21.2 (2021): 1090-1095; Loh, Joel YY, et al. "Near-perfect absorbing copper metamaterial for solar fuel generation." Nano Letters 21.21 (2021): 9124-9130; Liang, Yu, et al. "Polarization-controlled triple-band absorption in all-metal nanostructures with magnetic dipoles and anapole responses." Applied Physics Express 12.6 (2019): 062014.]
3. I have observed that your design's unit cell period (P) is 2um, while the operating wavelength range is around 0.3-0.8um. This means that P>λ. In this case, can your design be considered a "metasurface," which typically requires a sub-λ period size? Perhaps the term "meta-devices" would be more appropriate. However, I will leave this decision up to the authors.
readable
Reviewer 2 Report
The paper is well written and explained in proper manner. The reader can easily understand the finding. The graphs are of high quality and the study is well backed by the mathematical justification. Except minor typos and the formatting error there is no issue found in the manuscript and thus it can be accepted.
Minor typos are found at few places.
Reviewer 3 Report
Ana Barar et al., “Towards high-efficiency photon trapping in thin-film perovskite solar cells using etched fractal metasurfaces”, demonstrated this article under certain configurations of the architecture a thin-film perovskite solar cell, reflection values below 0.1 are obtained thoughout the visible domain. However, before any decision is made on its publication, mandatory revision is required in order to clarify some points and increase its attractiveness to the general public journal Materials. See comments below.
1. I would advise to pay attention to photovoltaic cells in order to increase the relevance of the photovoltaic industry. So the paper gives examples of the rapid growth of photovoltaic power, but I would like to see information about the maximum efficiency and about the stability problems of solar panels. Please pay attention to the following works (https://doi.org/10.3390/cryst12050699, https://doi.org/10.3390/ma15228151, https://doi.org/10.3390/molecules28031288).
2. In the section "Design considerations and simulation condition" you write "The considered structures are simulated and investigated using a commercially available Finite Element Time Domain (FETD)-based simulation software". Please explain which simulation application you used (maybe Ansys, Comsol Multiphysics, CST Studio, etc.), since there is little information to evaluate the simulation results. It is also necessary to describe the modeling conditions in more detail.
3. What is the reason for the choice of the presented height (he=0.2 µm)? Was there a profile height analysis ?
4. Coating layer height is exactly 0.2 µm ? maybe hc=2 µm? (Please take a closer look at Table 1.)
5. Please explain what "rounded profile" means. What is the radius of the rounding, what radii have been selected, etc.
6. There is no reference to Figure 3 in the text.
7. The paper considers 3 different configurations of the etched fractal metasurface. The optimal structure is the most developed surface "3C", but have you considered a more developed surface?
Moderate editing of English language
Round 2
Reviewer 3 Report
Accept in present form. All comments have been taken into account. The paper is well written and explained in proper manner.
Minor editing of English language required